# Real-time Breast Lesion Detection in Videos via Spatial-temporal Feature Aggregation

**Chao Qin**[1]  (iD)                                              CHAO.QIN@MBZUAI.AC.AE
[1] *Department of Computer Vision, Mohamed bin Zayed University of Artificial Intelligence, United Arab Emirates*

**Jiale Cao**[2]                                                  CONNOR@TJU.EDU.CN
[2] *School of Electrical and Information Engineering, Tianjin University, China*

**Fahad Shahbaz Khan**[1]                                         FAHAD.KHAN@MBZUAI.AC.AE
**Salman Khan**[1]                                               SALMAN.KHAN@MBZUAI.AC.AE
**Huazhu Fu**[3]                                                        HZFU@IEEE.ORG
[3] *Institute of High Performance Computing, Agency for Science, Technology and Research, Singapore*

**Ehud Ahissar**[4]                                         EHUD.AHISSAR@WEIZMANN.AC.IL
[4] *Department of Brain Sciences, Weizmann Institute of Science, Israel*

**Rao Muhammad Anwer**[1]                                       RAO.ANWER@MBZUAI.AC.AE

**Editors:** Accepted for publication at MIDL 2025

## Abstract

Recently, transformer-based detectors have shown impressive performance for breast lesion detection in ultrasound videos. However, these methods often require substantial computational resource and exhibit low inference speed, which poses challenges towards real-time applicability. To address this issue, we introduce a fast yet accurate spatial-temporal transformer, named FA-DETR, to efficiently aggregate multi-scale spatial-temporal features for breast lesion detection in ultrasound videos. Our FA-DETR is based on a lightweight spatial-temporal self-attention module, which seamlessly fuses spatial and temporal features extracted from each video frame. In the decoding phase, we employ IoU-aware query selection to generate independent queries for each frame. These queries gain access to rich spatial-temporal information through the encoder embeddings' cross-attention and frame-aware cross-attention mechanisms. Experiments conducted on a public breast lesion ultrasound video dataset demonstrate that our FA-DETR achieves state-of-the-art performance with an absolute gain of 3.8% in terms of overall AP while being 2.5 times faster, compared to the best existing approach in the literature. Our code is available at https://github.com/AlfredQin/FA-DETR.

**Keywords:** breast lesion, ultrasound video, real-time detection.

## 1. Introduction

Accurately detecting breast lesions in ultrasound videos is advantageous with the potential of significantly enhancing the efficiency of radiologists in diagnosing breast cancer through ultrasound imaging (Zhang et al., 2020; Movahedi et al., 2020; Qi et al., 2019; Xue et al., 2021; Yang et al., 2020; Yap et al., 2017; Zhu et al., 2020; Xie et al., 2020; Nie et al., 2019). Recent advances (Lin et al., 2022b; Qin et al., 2023; Yu et al., 2023) have been made due to transformer-based detectors that leverage temporal information in video-based

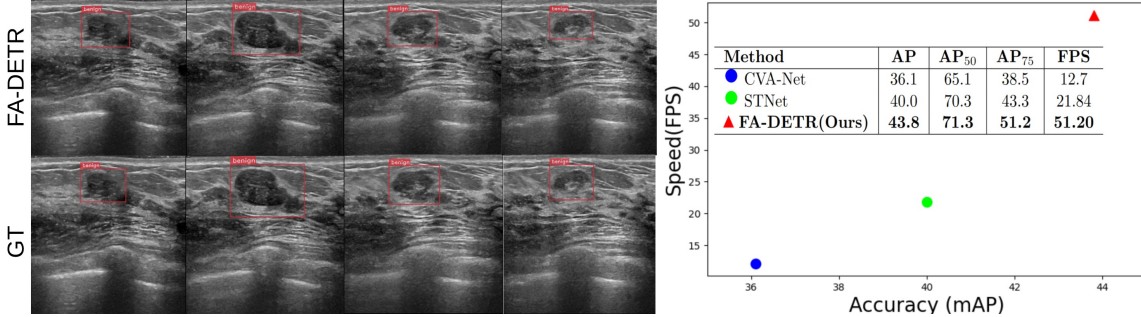

Figure 1: **On the left**: Qualitative results of our FA-DETR for breast lesion detection on example video frames. We also show ground-truth (GT). Our FA-DETR accurately localizes the lesion regions and correctly classifies the breast lesion. Additional results are presented in Fig.3. **On the right**: Comparison in terms of Accuracy (mAP) vs. Inference Speed (FPS) with existing methods. We also report performance at $AP_{50}$ and $AP_{75}$. Compared to the recent STNet (Qin et al., 2023), our FA-DETR achieves an absolute gain of 3.8% in terms of mAP, while being twice as fast at inference and operating at more than 50 frames per second (FPS) on an NVIDIA 3090 machine.

ultrasound detection. Yu *et al.* (Yu et al., 2023) utilize negative temporal contexts of previous frames to suppress the false positives in ultrasound video detection. The CVA-Net (Lin et al., 2022b) develop a feature aggregation network, which performs intra and inter-video fusions at video and clip level through attention blocks. Recently, STNet (Qin et al., 2023) introduce a spatial-temporal deformable attention module, enabling deep feature aggregation at every stage of both encoder and decoder. However, these recent methods are typically computationally intensive with slower inference speed, limiting their real-time practical applicability.

We distinguish that the multi-scale spatial-temporal attention mechanism within the encoder of recent STNet (Qin et al., 2023) poses a significant computational bottleneck, leading to redundancy. Further, we argue that the process of query initialization and interaction between the query and encoder features is sub-optimal, which undermines the overall performance. In this work, we address the combined challenge of designing a breast lesion detector for ultrasound videos that is *both* accurate and fast, thereby operating at real-time speed.

To tackle the aforementioned challenges, we introduce a fast yet accurate spatial-temporal transformer-based framework, named FA-DETR.

In the encoder of our FA-DETR, we substantially reduce the computational overhead by leveraging high-level features extracted from backbone network to perform temporal information fusion across different frames. Additionally, we employ a CNN-based approach for effective fusion of multi-scale features, instead of adopting the compute intensive transformer-based technique as in prior works. Contrary to STNet (Qin et al., 2023) where a global set of learnable queries is used for all frames, our FA-DETR sets a distinct set of queries for each frame image and initializes it with the top-K features output from the corresponding encoder. These queries first aggregate multi-scale spatial-temporal features across all frames and then interact with each other through a carefully designed multi-level spatial-

temporal decoder. We conduct experiments on publicly available breast lesion ultrasound video dataset, BLUVD-186 (Lin et al., 2022b). Results show the effectiveness of FA-DETR in terms of *both* efficiency and accuracy. Our FA-DETR achieves a mean average precision (mAP) of 43.8% with an absolute gain of 3.8% in terms of detection accuracy while being twice as fast (see Fig.1), compared to STNet (Qin et al., 2023).

## 2. Method

Here, we outline the structure of our FA-DETR, designed for the detection of breast lesions in ultrasound videos, in Fig. 2(a). FA-DETR processes six consecutive frames, $F_{t-2}$, $F_{t-1}$, $F_t$, $F_{t+1}$, $F_{t+2}$, and $F_{t+3}$, from an ultrasound video sequence. The first step involves utilizing a backbone network to extract deep multi-scale features from each frame individually. Afterwards, we employ a lightweight spatial-temporal encoder (LWST-Encoder) that efficiently executes temporal fusion of inter-frame features and intra-frame multi-scale feature fusion. Subsequently, a multi-level spatial-temporal deformable attention-based decoder (MLST-Decoder) is introduced to facilitate progressive interaction between the queries of the decoder and the temporal multi-scale features produced by the LWST-Encoder. The decoder's final output is then fed to a classifier and a bounding-box predictor, which are responsible for classification and bounding-box regression tasks, respectively. Specifically, the classifier consists of a linear projection layer, while the bounding box regression head employs a 3-layer feed-forward neural network (FFN) and a linear projection layer to predict bounding box coordinates.

### 2.1. Lightweight Spatial-Temporal Encoder

Recent studies (Lin et al., 2022a; Lv et al., 2023) have identified computational redundancies in the multi-scale transformer encoder of Deformable-DETR (Zhu et al., 2021), highlighting a significant increase in computational burden with minimal performance gains. This insight has led us to reconsider the necessity of applying multi-scale spatial-temporal attention to low-level features in STNet, given that these features often lack the rich semantic information required to effectively establish temporal connections between frames. Instead, we posit that high-level features, which contain richer contextual knowledge, are better suited for modeling these connections.

   To address this, we introduce the Lightweight Spatial-Temporal Encoder (LWST-Encoder), designed to segregate spatial-temporal multi-scale feature learning into two distinct processes: temporal information fusion and spatial multi-scale feature fusion. This approach is visualized in Fig. 2(b). Let $\left\{C_3^t, C_4^t, C_5^t\right\}$ denote the set of multi-scale feature maps at frame $t$. We begin by applying self-attention to the high-level feature $C_5^t$ for each frame individually, followed by a feed-forward network, resulting in the refined feature $D_5^t$. Subsequently, $D_5^t$ is enabled to aggregate temporal information by interacting with the refined high-level features of adjacent frames $\left\{D_5^{t-2}, D_5^{t-1}, D_5^{t+1}, D_5^{t+2}, D_5^{t+3}\right\}$ through a multi-head cross-attention block. The output feature from this temporal cross-attention block is enriched with temporal and contextual information, which is then used for inter-scale fusion with lower-level features. For this fusion process, we utilize a top-down Feature Pyramid Network (FPN) (Lin et al., 2017a) and a bottom-up Path Aggregation Network (PAN) (Liu et al., 2018), similar to RT-DETR (Lv et al., 2023) and PANet (Liu et al., 2018). This is

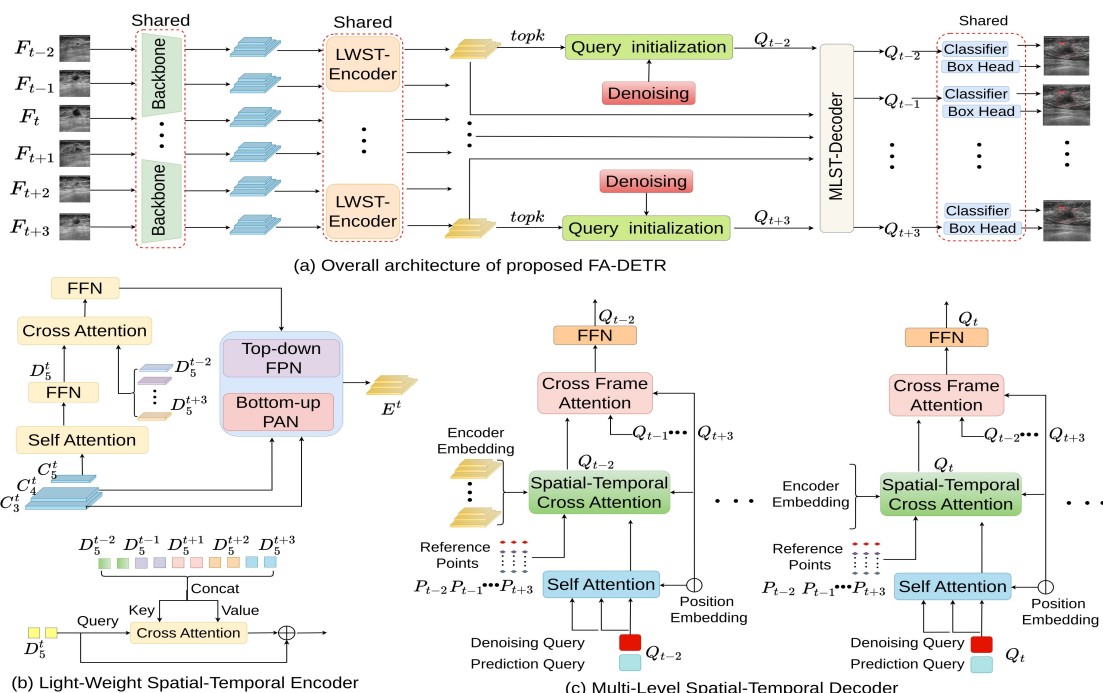

(a) Overall architecture of proposed FA-DETR

(b) Light-Weight Spatial-Temporal Encoder

(c) Multi-Level Spatial-Temporal Decoder

Figure 2: **(a)** Overall architecture of our FA-DETR. The proposed FA-DETR takes six frames as inputs and extracts multi-scale features of each frame. Afterwards, the proposed FA-DETR utilizes a lightweight spatial-temporal encoder (LWST-Encoder)**(b)** and multi-level spatial-temporal decoder **(c)** for spatial-temporal multi-scale information fusion. Finally, the proposed FA-DETR performs classification and regression. Specifically, sub-figure (b) consists of two parts, with the lower part providing a zoomed-in view of the cross-attention module shown in the upper part.

a significant departure from previous methods like STNet and CVA-Net, which employ up to six layers of deformable attention in the encoder, resulting in a heavier computational load. In contrast, our proposed encoder leverages a single layer of attention, significantly reducing computational costs. The deep multi-scale features $E_{t-2}$, $E_{t-1}$, $E_t$, $E_{t+1}$, $E_{t+2}$, $E_{t+3}$ are then forwarded to the proposed decoder for further processing.

## 2.2. Multi-Level Spatial-Temporal Decoder

Recent advancements (Liu et al., 2022; Yao et al., 2021; Wang et al., 2022; Zhang et al., 2022) in DETR have introduced the concept of interpreting the queries in the decoder as content and position queries. These studies propose various methods for selecting the top K features from the encoder, including those based on classification scores and Intersection over Union (IoU) scores. Drawing inspiration from this body of work, we posit that adopting learnable queries, as done in STNet, is suboptimal and impair the detector's performance. Furthermore, STNet utilizes a set of global queries across all frames. Given that STNet's decoder operates with a singular set of queries, the reference points for each frame during the cross-attention phase between the query and encoder embeddings remain unchanged.

We argue that this approach of using global queries and identical reference points adversely affects the accuracy of bounding box localization across frames, as the position and size of objects can vary significantly between frames.

To remedy this, we introduce a Multi-Level Spatial-Temporal Decoder (MLST-Decoder), depicted in Fig. 2(c). The MLST-Decoder comprises six blocks, each featuring multi-head self-attention, multi-scale spatial-temporal cross-attention, and multi-head cross-frame attention. In our decoder block, each frame is allocated an independent set of queries, $Q_t \in R^{N \times D}$. For streamline the training process, we adopt a strategy inspired by DINO (Zhang et al., 2022) and RT-DETR (Lv et al., 2023), constructing denoising queries based on ground truth and concatenating them with the prediction query. The denoising query is utilized only during training. These prediction queries are initialized with the top K encoder features, determined by classification and IoU scores.

As the queries $Q_t$ undergo self-attention, they assimilate multi-scale spatial-temporal features $E_{t-2}$, $E_{t-1}$, $E_t$, $E_{t+1}$, $E_{t+2}$, $E_{t+3}$ from the encoder, employing spatial-temporal multi-scale deformable cross-attention. In the $k$-th decoder layer's multi-scale spatial-temporal deformable cross-attention, the query $Q_t$ requires reference points predicted by the preceding $k-1$-th decoder layer to sample elements from the encoder features. Using the queries $Q_t^k \in R^{N \times D}$ in the $k$-th decoder layer at frame $t$ as an example, STNet employs identical reference points for all six frames.

In contrast to STNet, our proposed MLST-Decoder adopts a more dynamic approach for reference points, formulated as:

$$_{t+i}p_x^k =_{t+i} b_x^{k-1} +_{t+i} \Delta p_x \cdot_{t+i} b_w^{k-1}, \tag{1}$$

$$_{t+i}p_y^k =_{t+i} b_y^{k-1} +_{t+i} \Delta p_y \cdot_{t+i} b_h^{k-1}, \tag{2}$$

where reference points can dynamically adjust per frame, incorporating specific frame reference information. This method ensures that location of sampling points on reference frames is more accurately aligned with variable positions and sizes of objects across frames, enhancing the precision of bounding box predictions. Following interaction between queries and encoder features, we implement a cross-frame multi-head attention mechanism to facilitate communication between queries across different frames, further refining the detection process.

## 3. Experiments

### 3.1. Dataset and Implementation Details

**Dataset** We conduct our experiments using the publicly available BLUVD-186 dataset (Lin et al., 2022b), which was acquired by PHILIPS TIS L9-3 and LOGIQ-E9 machines. The dataset has 186 ultrasound videos, including 112 malignant, 74 benign cases, and 25,458 ultrasound frames. For consistency in comparison, we utilize the same training and test dataset splits as previous studies (Lin et al., 2022b; Qin et al., 2023).

**Evaluation Metrics** To evaluate the performance of breast lesion detection methods on ultrasound videos, we utilize three metrics: average precision (AP), $AP_{50}$, and $AP_{75}$. These metrics provide a comprehensive measure of the detection accuracy at different IoU thresholds. For frames near the boundaries (e.g, $t = 0$ or $t = T$), we pad the sequence by duplicating boundary frames to maintain a consistent input length of six frames.

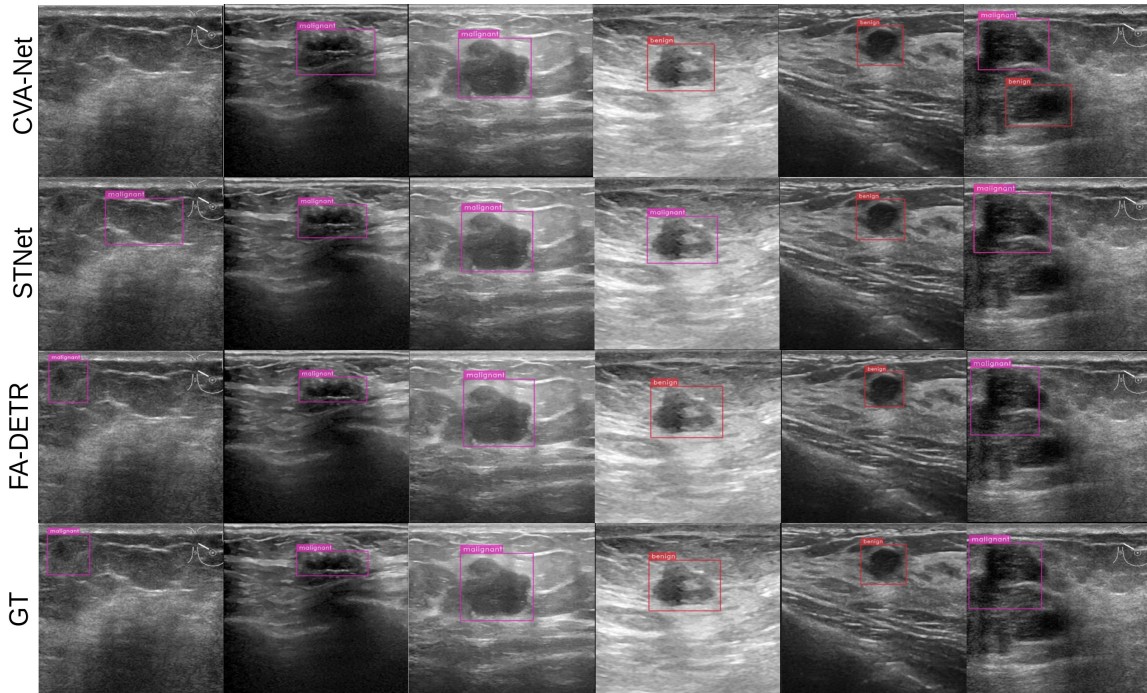

Figure 3: Qualitative breast lesion detection comparison on example ultrasound video frames between the CVA-Net (Lin et al., 2022b), STNet (Qin et al., 2023) and our proposed FA-DETR. We also show the ground truth as reference. Our FA-DETR achieves better detection performance, compared to STNet. Best viewed zoomed in.

**Frame Selection** We observe only marginal performance differences across various combinations of past and future frames with a total of six input frames, including the case where only past frames are used. Thus, we select a general temporal context ($F_{t-2}$ to $F_{t+3}$) in our manuscript to clearly illustrate our model's capability to utilize both past and future frames. **Implementation Details** Our model leverages the ResNet-50 (He et al., 2016) architecture, pre-trained on the ImageNet (Deng et al., 2009), as its backbone. To augment the diversity of the training data, we apply random flipping, cropping, resizing, and color jittering to the videos. The model is optimized using the AdamW optimizer, with a learning rate of $5 \times 10^{-5}$ and a weight decay of $1 \times 10^{-4}$. Training is conducted over 10 epochs on 4 NVIDIA 3090 GPUs, ensuring efficient utilization of computational resources for model development.

## 3.2. Comparison with State-of-the-Art

Our proposed approach is compared with twelve state-of-the-art methods, which include both image-based and video-based approaches. We report the detection performance of these methods except STNet as generated by CVA-Net. Specifically, CVA-Net obtains the detection performance of these methods by utilizing their publicly available codes or by re-implementing them in cases where no publicly available codes exist.

Table 1: State-of-the-art quantitative comparison of our approach with existing methods in literature on the BLUVD-186 dataset. Our approach achieves a superior performance on three different metrics. Compared to the recent STNet (Qin et al., 2023), our approach obtains a gain of 3.8% in terms of overall AP. We show the best results in bold.

| Method | Type | Backbone | AP | $AP_{50}$ | $AP_{75}$ |
|---|---|---|---|---|---|
| GFL (Li et al., 2020) | image | ResNet-50 | 23.4 | 46.3 | 22.2 |
| Cascade RPN (Vu et al., 2019) | image | ResNet-50 | 24.8 | 42.4 | 27.3 |
| Faster R-CNN (Ren et al., 2015) | image | ResNet-50 | 25.2 | 49.2 | 22.3 |
| VFNet (Zhang et al., 2021) | image | ResNet-50 | 28.0 | 47.1 | 31.0 |
| RetinaNet (Lin et al., 2017b) | image | ResNet-50 | 29.5 | 50.4 | 32.4 |
| DFF (Zhu et al., 2017b) | video | ResNet-50 | 25.8 | 48.5 | 25.1 |
| FGFA (Zhu et al., 2017a) | video | ResNet-50 | 26.1 | 49.7 | 27.0 |
| SELSA (Wu et al., 2019) | video | ResNet-50 | 26.4 | 45.6 | 29.6 |
| Temporal ROI Align (Gong et al., 2021) | video | ResNet-50 | 29.0 | 49.9 | 33.1 |
| MEGA (Chen et al., 2020) | video | ResNet-50 | 32.3 | 57.2 | 35.7 |
| CVA-Net (Lin et al., 2022b) | video | ResNet-50 | 36.1 | 65.1 | 38.5 |
| STNet (Qin et al., 2023) | video | ResNet-50 | 40.0 | 70.3 | 43.3 |
| **FA-DETR (Ours)** | video | ResNet-50 | **43.8** | **71.3** | **51.2** |

**Quantitative Comparisons** Table 1 offers a state-of-the-art quantitative comparison between our FA-DETR approach and twelve existing methods for breast lesion video detection in the literature. Generally, video-based methods are observed to achieve higher average precision (AP), AP50, and AP75 scores compared to those that are image-based. Among these methods, the recent STNet stands out with an impressive AP score of 40.0, AP50 score of 70.3, and AP75 score of 43.3. Our FA-DETR method surpasses STNet on all metrics, marking a notable advancement in breast lesion detection in ultrasound videos. Specifically, FA-DETR elevates the overall AP score significantly from 40.0 to 43.8, the AP50 score from 70.3 to 71.3, and the AP75 score from 43.3 to 51.2. This substantial improvement underscores the effectiveness of our proposed method in enhancing the detection accuracy of breast lesions in ultrasound video analysis.

**Qualitative Comparisons** Fig. 3 provides a qualitative comparison of breast lesion detection among CVA-Net, STNet, and our FA-DETR on an ultrasound video. In the first row, CVA-Net is shown to fail in identifying breast lesions in the first frame and falsely predicts a non-existent benign lesion in the last frame. While STNet performs better than CVA-Net, it also encounters misclassification issues in the fourth frame. In contrast, our FA-DETR method, depicted in the third row, successfully detects breast lesions in all frames with accurate classification and precise bounding box regression for each frame. This demonstrates the superior performance of our approach in accurately identifying and classifying breast lesions in ultrasound videos.

**Inference Speed Comparison** Fig. 1 showcases the inference speed of our FA-DETR compared to CVA-Net and STNet, evaluated on an NVIDIA RTX 3090 GPU under the same experimental conditions. The performance metric used for this comparison is FPS (frames

Table 2: Ablation study with different design choices. Our proposed FA-DETR achieves a superior performance compared to the baseline and some different designs. We show the est results in bold.

| Study | LWST-Encoder | MLST-Decoder | AP | $AP_{50}$ | $AP_{75}$ |
|---|---|---|---|---|---|
| Baseline | - | - | 32.7 | 53.1 | 38.3 |
| Baseline + LWST-Encoder | ✓ | - | 37.6 | 63.6 | 44.1 |
| Baseline + MLST-Decoder | - | ✓ | 38.8 | 65.8 | 45.4 |
| **FA-DETR (Ours)** | ✓ | ✓ | **43.8** | **71.3** | **51.2** |

per second). Our FA-DETR demonstrates superior efficiency with an average inference speed of 51.2 FPS. In comparison, CVA-Net achieves an average speed of 12.17 FPS, and STNet operates at 21.84 FPS. This indicates that our model is approximately 2.5 times faster than STNet, a significant improvement that can be attributed to the implementation of a lightweight spatial-temporal encoder in our FA-DETR. This enhancement in inference speed does not compromise the model's accuracy or detection performance, illustrating the effectiveness of our approach in providing fast and reliable breast lesion detection in ultrasound videos.

### 3.3. Ablation Study

To validate the effectiveness of our proposed lightweight spatial-temporal encoder and multi-level spatial-temporal decoder, we conducted a series of ablation studies. We established a baseline network by omitting the proposed lightweight spatial-temporal encoder from the FA-DETR and substituting its multi-level spatial-temporal decoder with the multi-scale decoder of the standard Deformable-DETR, which utilizes learnable queries without incorporating denoising queries. This baseline network extracts multi-scale features via the backbone and directly feeds these features into the multi-scale deformable DETR decoder, without any form of spatial-temporal fusion in both the encoder and decoder stages. The second network incorporates our proposed lightweight spatial-temporal encoder atop the baseline, maintaining the rest of the structure unchanged. In the third study, we introduced the proposed multi-level spatial-temporal decoder to the baseline network, omitting spatial-temporal fusion in the encoder. The fourth study features our fully proposed FA-DETR, equipped with both the lightweight spatial-temporal encoder and multi-level spatial-temporal decoder. The results, as presented in Table 2, demonstrate that incorporating the LWST-Encoder and MLST-Decoder improves the AP by 4.9 and 6.1, respectively, compared to the baseline network. This highlights the effectiveness of our approach in conducting inter-frame temporal feature fusion and intra-frame multi-scale feature fusion, even when only partially implemented in either the encoder or decoder. Furthermore, our complete FA-DETR model boosts the AP by 6.2 and 5.0 compared to using only the LWST-Encoder and MLST-Decoder, respectively. This suggests that the comprehensive integration of our lightweight spatial-temporal encoder and multi-level spatial-temporal decoder is crucial for aggregating multi-scale spatial-temporal features and achieving optimal detection performance. The synergy between these components significantly enhances breast lesion detection in ultrasound videos.

## 4. Conclusion

We introduce FA-DETR, a novel real-time approach for breast lesion detection in ultrasound videos, featuring a lightweight encoder that effectively fuses temporal multi-scale features extracted from the backbone. Additionally, our multi-level spatial-temporal decoder aggregates deep multi-scale features, enhancing the model's detection capabilities. Due to the efficient design of our encoder and the multi-level interaction of queries in the decoder, FA-DETR accelerates inference speed while achieving superior detection performance. Experiments on a public breast lesion ultrasound video dataset demonstrate the effectiveness of the proposed FA-DETR.

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
