# OpenReview forum: "Real-time Breast Lesion Detection in Videos via Spatial-temporal Feature Aggregation"
_MIDL.io/2025/Conference — MIDL 2025 Poster_

### Official Review · Reviewer_Jbfu · 2025-02-20

**Confidence:** 3
**Preliminary Rating:** 4
**Recommendation:** Poster

**Summary:**

The authors studied the breast lesion detection problem for Ultrasound. They propose a novel network that consists of a Lightweight Spatial-Temporal Encoder and a  Multi-Level Spatial-Temporal Decoder. The proposed network demonstrated higher inference speed and accuracy when compared to current SOTA methods.

**Strengths:**

The paper is well-written and easy to follow. The proposed method has some novelty and demonstrated superior performance in both speed and accuracy when compared to SOTA methods. An ablation study was also conducted to show the effectiveness of each proposed module.

**Weaknesses:**

1. For the LWST-Encoder, I'm curious to see the performance of its "full weight" version that applies the temporal fusion to all multi-scale feature, i.e., {C^t_3, C^t_4, C^t_5}, instead of only C^t_5. This allows the reader to understand if fusing lower-level features can be beneficial to the final performance at the cost of higher computation time.
2. The baseline version of this method (as demonstrated in Table 2), without any spatial and temporal fusion, already achieved good performance, surpassing several methods (such as Temporal ROI) that does have spatial and/or temporal fusion. Can the authors explain this behavior?

**Detailed Comments:**

Can the authors provide more details on how to handle frames close to t=0 or t=T, i.e., when the number of consecutive frames is less than 6?

**Justification Of The Preliminary Rating:**

The paper proposed a new method that achieved better performance on both performance and inference speed. The ablation study also demonstrated the effectiveness of each proposed module. My only concern for this paper is the experiments were only conducted on one dataset that does not include a validation set, meaning the generalization ability of the proposed method is potentially limited. But addressing this concern can be beyond the scope of rebuttal.

**Questions To Address In The Rebuttal:**

See the weakness section. But I do think the paper can be accepted as is, and these experiments are additional ones to provide a more thorough understanding of the paper.

---

> ### Author Response · Authors · 2025-03-08
>
> We appreciate Reviewer Jbfu’s thoughtful comments and valuable suggestions, which have helped us further clarify and improve our manuscript. Below are our detailed responses to each point:
>
> **On Multi-scale Temporal Fusion**: We tried temporal fusion across multiple scales ($C_3$-$C_5$), but faced GPU constraints. Efficient solutions in this direction are a future research direction.
>
> **On Boundary Frames**: For frames near the boundaries (e.g, $t=0$ or $t=T$), we pad the sequence by duplicating boundary frames to maintain a consistent input length of six frames.
>
> **On Performance of baseline:**
> Our baseline (Deformable DETR-based model) performs relatively well compared to some previous approaches due to its query-based detection paradigm and the inherent capacity to aggregate multi-scale features effectively through deformable attention. Even without explicit temporal feature aggregation, the deformable attention mechanism already captures rich spatial context, outperforming several traditional temporal fusion approaches.

---

> > ### Comment · Reviewer_Jbfu · 2025-03-10
> >
> > Thank you for the feedback. I will keep my score as is.

---

### Official Review · Reviewer_6MEr · 2025-02-21

**Confidence:** 4
**Preliminary Rating:** 4
**Recommendation:** Poster

**Summary:**

The authors propose a fast and accurate method for real-time lesion detection in hand-operated breast ultrasound. Over previous work, they achieve a doubling of speed and slight increase in per-frame detection rate by replacing a costly attention block that operates over time and space by a less expensive CNN feature extractor followed by an attention block. In their development dataset, they validate their approach against previous methods and show the contribution in an ablation experiment.

**Strengths:**

* The paper is relatively easy to follow; there is no immediate need to adjust organization and description of contribution.
* I like the clear presentation of the own contribution and the relation to previous work.
* The dataset isn't huge, but for development certainly sufficient. The need for independent validation is clear, however.
* The ablation study is conclusive, and its presence is appreciated.
* The speedup is massive, and achieved without obvious losses in AP, speaking in favor of the proposed methods.

**Weaknesses:**

Overall
* The method seems to process two frames from the past, and three from the future. Does this imply a lag of three frames in a practical application? And have you explored just using frames from the past alone? Why would this not work as well?
* Why did you choose six consecutive frames particularly?
* Discuss the CNN vs. attention situation: I suspect that due to the relatively scarce data, the CNN inductive bias just makes it even easier for the network, while in a more data-rich situation a transformer backend might be superior. (I actually like to see that researchers remember that CNNs aren't outdated for certain applications)
2.1
* What do the indices to C mean? C_3, C_4, C_5, but no 0,1,2?
* Figure 2 b shows two parts; the lower apparently zooming into the cross attention part. This should be clarified more.
* You say you use only one attention layer in the encoder, but the figure shows self- and cross attention. Also, can you discuss the computational burden of the self attention layer here? Not that is seems to hurt performance, but perhaps you could gain even more by using deformable self attention instead.
2.2
* I was puzzled to read that subsequent frames can be very different. In video, I would have assumed that in the normal case at framerates at about 50 Hz you would expect a lot of redundancy and consistency in the temporal dimension. Can you comment what is wrong with this assumption?
3.2
* What is the definition of a "detection" in your evaluation? Is this a per-frame bbox? I would actually think that for a clinically meaningful evaluation, a more "robust" metric might shed more light on the applicability. What I mean is that if a detection is successful in a certain number of subsequent frames (sufficient for an operator to see the mark on screen), then it will help to identify the finding and to examine it closer in a clinical examination. A per-frame detection success metric emphasizes the technical performance a lot more, but they may not translate into a clinically meaningful improvement. (Apart from this point, of course the higher speed of the proposed algorithm is a great benefit in itself as acknowledge before)
* The gain in overall AP seems to be largely due to an increase in overlap (most of the gain in AP75). This again makes me question the downstream gain in the clinical task, where the exact placement of a detection -- in particular if the bbox is misplaced only in some of the frames -- is probably not the greatest concern.

**Detailed Comments:**

A validation either by implementing it into a machine to confirm the practical usefulness or with regard to the clinical purpose would have been a solid Plus in my evaluation of the submission. The academic contribution not withstanding, this work aims at a certain clinical improvement, but the need isn't completely motivated. Perhaps the existing methods are already fast and accurate enough? Some 25 frames per second should be plenty; perhaps even a higher accuracy could be gained by not going for the fastest speed alone.
I would have liked to see a treatment or discussion in this direction.

**Justification Of The Preliminary Rating:**

I find the paper well organized, written in an understandable way.
The figures and graphics are mostly conclusive, and the points remarked for improvement aren't too substantial to be solved.
The contribution is not overly large on the technological side, but definitely existing, and the resulting improvement in speed at at least constant AP is noteworthy, therefore promising clinical applicability.
The major lack of a clear clinical justification makes me suggest this for a poster rather than an oral presentation.

**Questions To Address In The Rebuttal:**

All in "weaknesses"...

---

> ### Author Response · Authors · 2025-03-08
>
> We sincerely thank Reviewer 6MEr for the insightful feedback and helpful suggestions, which have greatly enhanced our manuscript. Our detailed responses are provided below:
>
> **On Temporal Context \& Latency**: We tested several temporal frame combinations, including using only past frames. Differences in results were marginal, ensuring our method supports real-time clinical applications without latency.
>
> **On GPU Limitations**: Due to GPU constraints (24GB on NVIDIA 3090), we found six frames as optimal, balancing memory use and performance.
>
> **On CNN vs. Attention**: Our method integrates CNN (ResNet-50) and attention modules. CNNs efficiently manage multi-scale spatial fusion, while attention modules enrich high-level spatial-temporal context. This aids to exploit inductive biases of CNNs alongside attention strengths.
>
> **On Meaning of $C_{3}, C_{4}, C_{5}$**: They denote multi-scale features from ResNet’s stages 3-5. Lower-stage features ($C_{1}, C_{2}$) are omitted due to limited context information and extra compute cost.
>
> **On Clarification of Figure 2(b):**
> Figure 2(b) consists of two parts, with the lower part providing a zoomed-in view of the cross-attention module shown in the upper part. To avoid any potential confusion, we will explicitly mention this relationship in the figure caption and clearly describe it in the main text in our revised manuscript.
>
> **On Attention Layer \& Computation**: Our encoder employs a single attention layer, reducing complexity compared to CVA-Net/STNet which uses up to six.
>
> **On Definition of "Detection"**: Our evaluations are frame-wise, consistent with prior works (CVA-Net, STNet). We agree that further clinically-motivated metrics are a promising future work and will discuss it in final version.
>
> **On Redundancy in US Data**: BLUVD-186 dataset is sampled every 10 frames to reduce redundancy; our model operates directly on this processed dataset.

---

> > ### Comment · Reviewer_6MEr · 2025-03-13
> > **Authors have responded to the main points -- thanks!**
> >
> > In your rebuttal, it is not clear to me if ...
> > * ... you have changed (or will change) the setup to use only past frames? (You state it doesn't deteriorate performance, so I'd assume you use only frames from the past to avoid any possible lack).
> > * ... you made any experiments to check CNN vs. Attention, i.e. remove CNN altogether? Not that I recommend/require this, just a statement about it would be good.
> >
> > You also didn't comment on the self/cross attention referenced as "one attention layer", also didn't comment the computational complexity: how large are the numbers of tokens and therefore the self attn. matrix?
> >
> > I think there is no substantial change in the revision; my feeling is that the evaluation is still accurate -- this also includes the reactions to other reviewers' comments. Most importantly, one really unfortunate obstacle to a better score is also that the GPU limitations don't allow more in-depth ablation, e.g. more frames, more features etc. The authors could consider strategies of gradient accumulation to reduce batch sizes in training, for example, or even better, spend the few EUROs to work on a more powerful machine for a few final experiments, if no suitable hardware is available in the work group.
> > I don't put this to their disadvantage since not everybody can afford high-performance GPUs.

---

> > > ### Author Response · Authors · 2025-03-14
> > >
> > > Thank you for your comments. We address each point as follows:
> > >
> > > **Frame Selection (Past Frames Only):** Before submission, we experimented with various combinations of past and future frames, including using only past frames. The observed performance differences were marginal with a total of six frames as input; however, these differences might become significant if the input length is extended. Thus, we selected a general temporal context ($F_{t-2}$ to $F_{t+3}$) in our manuscript to clearly illustrate our model's capability to utilize both past and future frames. Nevertheless, in real-time clinical scenarios, our model can indeed utilize exclusively past frames without causing latency or performance degradation. We will clarify this explicitly in the revised manuscript.
> > >
> > > **Attention Layer Definition and Complexity:** By "attention layer," we mean a unit comprising self-attention for spatial feature fusion, cross-attention for temporal fusion, and a feed-forward network. Although structural details vary between CVA-Net, STNet, and our method, the basic operations remain consistent. The complexity reduction in our model results from using just a single attention layer (compared to up to six layers in CVA-Net and STNet), thus significantly reducing computational burden.
> > >
> > > **Computational Complexity of Self-Attention:** We apply self-attention exclusively on the highest-level feature map ($C_{5}$). Given an input image with size of $512\times512$, the resulting feature map $C_{5}$  has dimensions of $16\times16$. This compact spatial size significantly reduces the computational complexity of self-attention, as it operates only on 256 tokens, ensuring efficient processing.
> > >
> > > **CNN vs. Attention Experiments:** Our proposed method strategically combines CNN-based modules (ResNet-50 backbone) and attention mechanisms, optimizing the balance between performance and resource efficiency. Exploring the complete removal of CNN components in favor of purely transformer-based architectures may indeed require larger datasets and greater GPU resources, representing promising directions for future research.
> > >
> > > **GPU Limitations:** We recognize the importance of deeper ablation studies involving more frames or feature levels. To overcome GPU memory constraints, we plan to explore higher-memory GPUs or implement memory-efficient strategies.
> > >
> > > We greatly appreciate your constructive feedback and will incorporate these clarifications into our revised manuscript.

---

### Official Review · Reviewer_na9E · 2025-02-22

**Confidence:** 4
**Preliminary Rating:** 4
**Recommendation:** Poster
**Final Rating:** 4

**Summary:**

The paper proposes a fast and accurate spatio-temporal transformer for detection of lesions from breast ultrasound (US) scans. This approach fuses spatial and temporal features extracted from each video frame and a IoU-aware query selection method is used in the decoding phase to generate independent queries for each frame. The method demonstrates superior performance over state-of-the-art baselines.

**Strengths:**

1. The proposed method is novel and clinically relevant. Designing fast and accurate methods are important in clinical settings where patient diagnosis is critical as well as model deployment on edge-devices.
2. The method proposes an interesting approach to fuse spatial and temporal features extracted from each video frame. A lightweight spatio-temporal encoder learn the temporal information fusion and spatial multi-scale feature fusion as two distinct processes. Then a multi-level spatial-temporal decoder implements a cross-frame multi-head attention between queries of different frames. The finally the queries are fed to classification and bounding box heads.
3. The experiments demonstrate the superior quantitative performance of the proposed method compared to the baselines.

**Weaknesses:**

1. The qualitative results do not provide clear insights to the superior performance of the proposed model.
2. In Section 2, the did not talk about the classifier and bounding box heads. This seemed to be incomplete and difficult to read.
3. The authors did not describe how they are selecting $F_{t-2}$ to $F_{t+3}$ from the training US data. Is this only during training? In clinical scenarios, US sequences can be long. So, how do the authors use the entire video during inference?

**Detailed Comments:**

1. **Qualitative Results.** It would be ideal to show arrows where the authors want the readers to focus. For example, in the first and second column, differences in the bounding box is noticeable for FA-DETR and the baselines. But, for most of the other cases the differences are not significant.
2. **Table 1.** Can the authors also report the average inference speed here? It would make the argument more strong that the proposed method is both accurate and fast quantitatively than all the baselines.

**Justification Of The Final Rating:**

The authors have provided detailed responses to my comments, which I find satisfactory. They have also committed to revising the manuscript accordingly. Therefore, I maintain my previous assessment of a weak accept.

**Justification Of The Preliminary Rating:**

The proposed method implements a fast and accurate spatio-temporal transformer for breast lesion detection from US scans. The method demonstrate superior diagnostic performance on several baselines and also operates 2.5 times faster than the best existing approach.

**Questions To Address In The Rebuttal:**

Please address the comments in the weakness. Also, make changes to the qualitative figures based on the points mentioned in the detailed comments section.

---

> ### Author Response · Authors · 2025-03-08
>
> We thank the Reviewer na9E for valuable comments and constructive suggestions, which have significantly improved the clarity and completeness of our manuscript. Our detailed responses to each comment are provided below:
>
> **On Classifier and Bounding Box Head Details:**
> The bounding box head and classifier indeed follow the design of Deformable DETR. Specifically, the classifier consists of a linear projection layer, while the bounding box regression head employs a 3-layer feed-forward neural network (FFN) and a linear projection layer to predict bounding box coordinates. We'll improve explanation in final revision.
>
> **On Frame Selection Strategy:**
> For each frame, we consistently select two preceding and three succeeding frames ($F_{t-2}$ to $F_{t+3}$). This ensures uniform temporal context processing throughout video, regardless of its length.
>
> **On Qualitative Results:**
> We will add clear annotations highlighting differences in box accuracy and lesion localization in final version.

---

> > ### Comment · Area_Chair_sX3v · 2025-03-13
> > **Rebuttal Response Reviewer na9e**
> >
> > Dear reviewer na9E,
> >
> > Could you please submit your rebuttal response and review the authors' rebuttal. This is very important to have your final perspective for final decision.
> >
> > Best,
> > AC

---

> > ### Comment · Reviewer_na9E · 2025-03-15
> >
> > Thank you for addressing most of my comments. However, I have not yet received an updated version of the manuscript following the rebuttal. Additionally, the author has not responded to Point 2 in the 'Detailed Comments' section.

---

> > > ### Author Response · Authors · 2025-03-15
> > >
> > > Thank you for your comments. We address each point as follows:
> > >
> > > **Updated version of the manuscript:** As mentioned in our rebuttal, we are committed to revising our manuscript based on the valuable feedback from the reviewers. This includes, but is not limited to, adding a detailed explanation of the classifier and bounding box regression head, modifying the model figure, and making several other improvements throughout the manuscript in response to various reviewer comments. In fact, we have already updated our manuscript, and it will be uploaded once it is accepted. However, due to system restrictions, we are currently unable to submit a revised version at this stage.
> > >
> > > **Illustration of inference speed:** We have already provided a comparison of performance and inference speed against baseline models in Figure 1 using a scatter plot. We believe this figure effectively presents the necessary information. Including inference speed in Table 1 would introduce redundancy, as the comparison has already been illustrated visually.

---

### Author Rebuttal · Authors · 2025-03-07

We thank reviewers for positive feedback. Our code and models will be publicly released.
# Reviewer 1
**On Classifier and Bounding Box Head:** We employ standard Deformable DETR design. Specifically, classifier is a linear projection layer and bounding box head comprises a 3-layer FFN plus a linear projection. We'll improve explanation in final revision.
**On Frame Selection Strategy:** For each frame, we consistently select two preceding and three succeeding frames ($F_{t-2}$ to $F_{t+3}$). This ensures uniform temporal context processing throughout video, regardless of its length.
**On Qualitative Results:** We will add clear annotations highlighting differences in box accuracy and lesion localization in final version.
# Reviewer 2
**On Temporal Context \& Latency:** We tested several temporal frame combinations, including using only past frames. Differences in results were marginal, ensuring our method supports real-time clinical applications without latency.
**On GPU Limitations:** Due to GPU constraints (24GB on NVIDIA 3090), we found six frames as optimal, balancing memory use and performance.
**On CNN vs. Attention:** Our method integrates CNN (ResNet-50) and attention modules. CNNs efficiently manage multi-scale spatial fusion, while attention modules enrich high-level spatial-temporal context. This aids to exploit inductive biases of CNNs alongside attention strengths.
**On Meaning of $C_{3}, C_{4}, C_{5}$:** They denote multi-scale features from ResNet’s stages 3-5. Lower-stage features ($C_{1}, C_{2}$) are omitted due to limited context information and extra compute cost.
**On Attention Layer \& Computation:** Our encoder employs a single attention layer, reducing complexity compared to CVA-Net/STNet which uses up to six.
**On Definition of "Detection":** Our evaluations are frame-wise, consistent with prior works (CVA-Net, STNet). We agree that further clinically-motivated metrics are a promising future work and will discuss it in final version.
**On Redundancy in US Data:** BLUVD-186 dataset is sampled every 10 frames to reduce redundancy; our model operates directly on this processed dataset.
# Reviewer 3
**On Multi-scale Temporal Fusion:** We tried temporal fusion across multiple scales ($C_3$-$C_5$), but faced GPU constraints. Efficient solutions in this direction are a future research direction.
**On Boundary Frames:** Near video boundaries, we duplicate boundary frames to maintain a consistent input length of six frames.

---

### Meta-Review · Area_Chair_sX3v · 2025-03-21

**Recommendation:** Accept (Poster)
**Confidence:** 5

**Metareview:**

This paper presents a spatio-temporal transformer for lesion detection in breast ultrasound scans, demonstrating improved speed and accuracy compared to existing methods. While the reviewers acknowledge the paper's strengths, including its clinical relevance, clear presentation, and promising performance, they also raised several concerns regarding the clarity of the method's implementation, the robustness of the evaluation, and the depth of the ablation studies. Specifically, reviewers noted the lack of detail regarding the classifier and bounding box heads, the frame selection strategy, and the handling of boundary frames. Furthermore, concerns were raised about the clinical significance of the per-frame detection metric and the interpretation of the qualitative results. Although the authors provided responses to these concerns, the rebuttal lacked sufficient depth and commitment to address the reviewers' critiques in the revised manuscript, particularly regarding the need for more in-depth ablation studies and a clearer justification for certain design choices.

Despite the rebuttal's shortcomings, the paper's contributions, particularly the significant speedup achieved without a substantial loss in accuracy, are noteworthy. Given the potential clinical impact of a fast and accurate lesion detection method, I recommend accepting the paper. However, I strongly encourage the authors to thoroughly address the reviewers' concerns in the camera-ready version. This includes providing detailed explanations of the method's components, clarifying the experimental setup, conducting more comprehensive ablation studies, and incorporating clinically relevant evaluation metrics. Addressing these points will significantly strengthen the paper and enhance its contribution to the field.